# A novel member of drug/metabolite transporter (DMT) family efflux pump, SA00565, contributes to tetracycline antibiotics resistance in *Staphylococcus aureus* USA300

Daiyu Li,[1] Yan Ge,[1] Ning Wang,[1] Yun Shi,[1] Gang Guo,[1] Jing Zhang,[1] Quanming Zou,[2] Qiang Liu[1]

**ABSTRACT** Drug efflux systems have recently been recognized as a significant mechanism responsible for multidrug resistance in bacteria. In this study, we described the identification and characterization of a new chromosomally encoded efflux pump (SA00565) in *Staphylococcus aureus*. SA00565, which belongs to the drug/metabolite transporter (DMT) superfamily, was predicted to be a 10-transmembrane segment transporter. To evaluate the role of *sa00565* in resistance, we generated *sa00565* gene deletion mutant (Δ*sa00565*) and assessed its susceptibility to 35 different antibiotic treatments. Our results demonstrated that the Δ*sa00565* mutant exhibited reduced resistance to tetracycline and doxycycline, with 64-fold and 12-fold decreased MICs, respectively. The mechanism of SA00565-mediated tetracycline resistance was demonstrated that SA00565 possesses the capability to efficiently extrud intracellular tetracycline into the environment. The efflux activity of SA00565 was further validated using EtBr accumulation and efflux assays. In summary, our study uncovered a previously unknown function of a DMT family transporter, which serves as a tetracycline efflux pump, thereby contributing to tetracycline resistance in *S. aureus*.

**IMPORTANCE** In this study, we addressed the significance of drug efflux systems in multidrug resistance of *Staphylococcus aureus*, focusing on the unexplored efflux pump SA00565 in the drug/metabolite transporter (DMT) superfamily. Through phylogenetic analysis, gene knockout, and overexpression experiments, we identified the role of SA00565 in antibiotic resistance. The Δ*sa00565* mutant showed increased susceptibility to tetracycline and doxycycline in disk diffusion assays, with significantly lower MICs compared to the WT. Remarkably, intracellular tetracycline concentration in the mutant was two- to threefold higher, indicating SA00565 actively eliminates intracellular tetracycline. Our findings emphasize the pivotal contribution of SA00565 to tetracycline antibiotic resistance in *S. aureus*, shedding light on its functional attributes within the DMT superfamily and providing valuable insights for combating multidrug resistance.

**KEYWORDS** *S. aureus*, drug efflux pump, antibiotic resistance, DMT transporters, tetracycline resistance

*S*taphylococcus aureus is a Gram-positive bacterial species responsible for causing bacteremia, endocarditis, skin and soft tissue infections, bone and joint infections, and hospital-acquired infections (1). Its reputation as a formidable pathogen is partly attributed to its proclivity for developing resistance to a broad range of antimicrobial agents (2). The resistance mechanisms employed by *S. aureus* are diverse and encompass several strategies. These include reducing the affinity of antibiotics by enzymatically modifying their binding sites, dismantling or neutralizing antimicrobial drugs enzymatically, decreasing the permeability of bacterial cells to antibiotics, as well as

Address correspondence to Qiang Liu, liuqiangdyy@163.com.

The authors declare no conflict of interest.

See the funding table on p. 9.

utilizing multidrug efflux pumps to extrude antimicrobial substances and thereby lower intracellular concentrations of antibiotics (3–8).

Multidrug efflux pumps are proteins that are embedded within bacterial membranes and facilitate the removal of harmful agents, such as antibiotics, biocides, and toxic metals, from the bacteria, exporting them into the surrounding environment (9). Collective consensus suggests that multidrug efflux pumps can be organized into five families: the small multidrug resistance (SMR) family, the major facilitator superfamily (MFS), the multidrug and toxic compound extrusion (MATE) family, the resistance-nodulation-cell division (RND) superfamily, and the adenosine-triphosphate (ATP)-binding cassette (ABC) superfamily (10). While ABC transporters rely on ATP, the other four families are secondary transporters that use an electrochemical gradient to move their substrates.

Beyond the mentioned secondary transporters, previous research has identified several other families of secondary active transporters, such as rhamnose transporter (RhaT), eukaryotic organellar triose phosphate transporter (TPT), and nucleotide-sugar transporter (NST) families (11). Taken together, these known and yet-to-be-discovered families constitute a ubiquitous superfamily that is widely distributed in bacteria, archaea, and eukaryotes. Within this superfamily, Donald et al. have specifically categorized these transporters as the drug/metabolite transporter (DMT) superfamily, comprising 14 distinct subfamilies (11).

To date, more than 15 efflux pumps in Staphylococci have been identified as contributing to antibiotic resistance in *S. aureus*, including NorA, LmrS, SdrM, MdeA, QacD, AbcA, and MepA (9). However, advancements in bioinformatics and genome analysis have led to the discovery of more than 30 putative efflux pumps in the *S. aureus* chromosome (12), with half of them having unknown functions. In this report, we present the identification and characterization of the *sa00565* gene, which encodes a novel transporter with 10 predicted transmembrane helices and belongs to the drug/metabolite exporter (DME) subfamily of DMT superfamily. We hypothesize that SA00565 serves as an efflux pump and examined its role in mediating antibiotic resistance in *S. aureus*.

## RESULTS

### Gene *sa00565* encodes a transporter closed to DME in DMT superfamily

In the *S. aureus* USA300_FPR3757 (accession number, NC_007793) genome, the gene with the locus tag of SAUSA300_00565 (named as *sa00565*) was annotated as encoding an EamA-like transporter with a molecular mass of 32.2 kDa. Our analysis using a transporter online database revealed that this gene encodes a 10 TMS transporter (Fig. 1A) that belongs to the DMT superfamily, a group that encompasses various membrane transporters classified into over 14 subfamilies (11). To elucidate the functional properties of SA00565, we undertook an analysis of its protein sequence alignment with various members of the DMT superfamily, and utilized this to construct an unrooted phylogenetic tree. Our analysis indicated that SA00565 shared a close affiliation with transporters grouped under the DME subfamily (Fig. 1B). These findings supported the possibility that the transporter gene is involved in the active efflux of drugs or metabolites from within *S. aureus*.

### Knockout of *sa00565* gene conferred *S. aureus* more sensitive to tetracycline and doxycycline

To determine the role of *sa00565* gene in mediating antibiotic resistance, a knockout mutant strain lacking the *sa00565* gene (Δ*sa00565*) as well as a complemented version of this strain with *sa00565* gene overexpressed in Δ*sa00565* (Δ*sa00565* + P$_{08825\_00565}$) were generated. The augmented expression of *sa00565* in the *sa00565* overexpression strain was authenticated by RT-qPCR, which manifests a 900-fold augmentation in its expression level when compared to the WT strain (Fig. S1). The growth of *S. aureus* in TSB

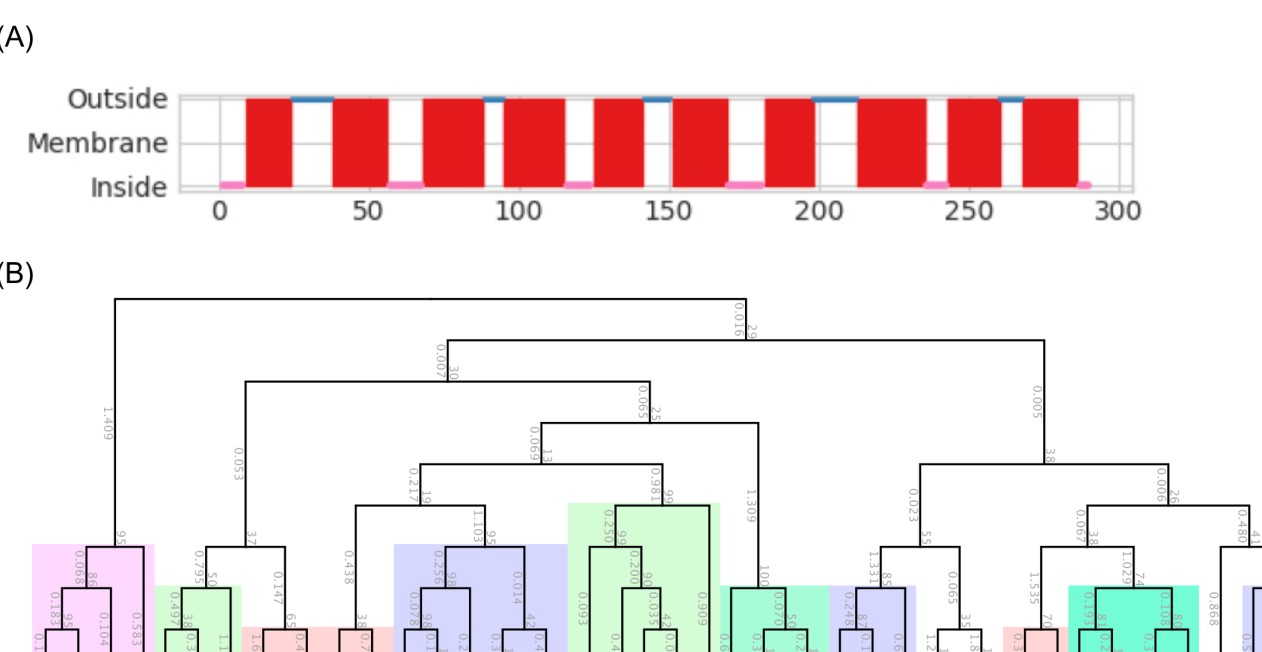

**FIG 1** Transmembrane topology and phylogenetic tree of SA00565 within the DMT superfamily. (A) Transmembrane topology of the predicted efflux pump SA00565. (B) The tree was generated by aligning the protein sequences of SA00565 and proteins from 14 classified groups of DMT transporters using ClustalW software in the CLC Main Workbench. The neighbor-joining method and Jukes-Cantor protein distance model were used to calculate the resulting tree. The abbreviated name of each classified subgroup in the DMT superfamily was listed in the Materials and Methods section, and SA00565 was highlighted with a red star.

medium remained unaltered following either knockout or overexpression of *sa00565* gene, as inferred from the growth curves of these strains (Fig. 2B).

Subsequently, the susceptibility of *S. aureus* USA300 WT, Δ*sa00565*, and *sa00565* overexpression strains to 35 different antibiotics was assessed through disk diffusion testing (Fig. S2). The results suggested that there were no statistically significant differences observed in the inhibitory zone diameters among the three strains for most of the antibiotics tested (Table S2). However, it was observed that tetracycline and doxycycline showed a more extensive inhibitory zone in the Δ*sa00565* mutant (27 mm by tetracycline and 27.5 mm by doxycycline) in comparison to the WT strain (10.5 mm by tetracycline and 15 mm by doxycycline). This increased susceptibility was reverted by the overexpression of *sa00565* in Δ*sa00565* mutant (Fig. 2A). The altered susceptibility of *S. aureus* to tetracycline, resulting from the knockout of the *sa00565* gene, was further substantiated through monitoring the growth of different strains in the presence of tetracycline in TSB liquid medium (Fig. 2B).

## Quantification of *sa00565*-conferred tetracycline and doxycycline resistance to *S. aureus*

To further validate and quantify the role played by *sa00565* in mediating tetracycline and doxycycline resistance in *S. aureus*, MICs assay for *S. aureus* WT, Δ*sa00565*, and Δ*sa00565* + P$_{08825\_00565}$ strains were performed. Our findings demonstrated a

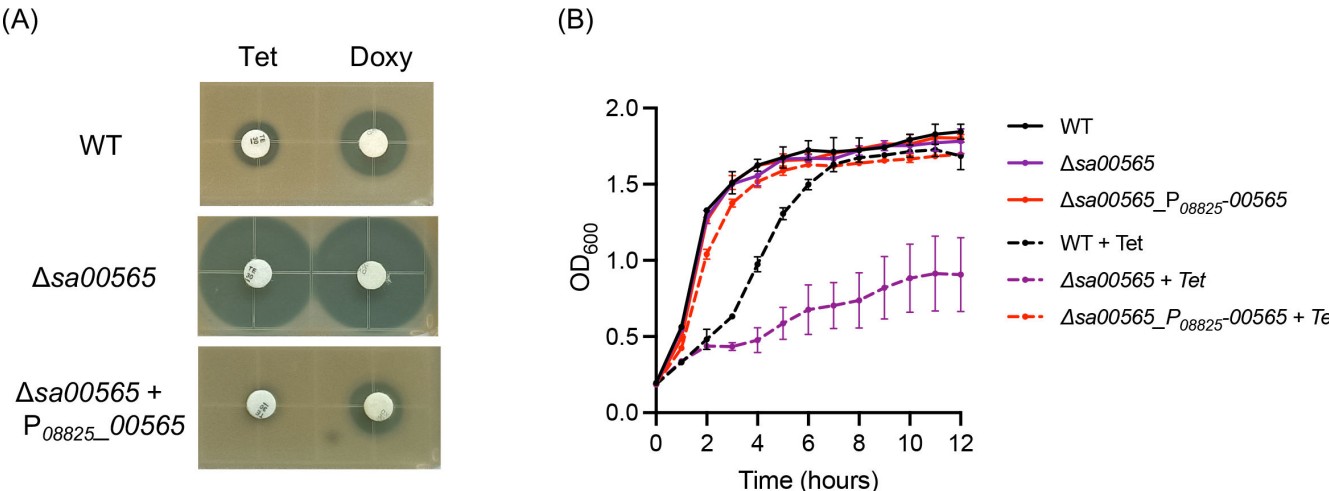

**FIG 2** (A) The susceptibility of *S. aureus* WT, Δ*sa00565*, and *sa00565* overexpression strains to tetracycline (Tet) and doxycycline (Doxy) in disk diffusion assay. (B) Growth curves of *S. aureus* USA300 WT, Δ*sa00565,* and *sa00565* overexpression strain in liquid TSB medium with (dotted line) or without (full line) 5 µg/mL tetracycline.

significant reduction in tetracycline MIC for the Δ*sa00565* strain (0.125 mg/L) as compared to the WT strain (8 mg/L), exhibiting up to a 64-fold decrease (Fig. 3). As expected, the *sa00565* overexpression strain displayed the highest MIC with 32 mg/L, which was fourfold higher than that of the WT strain. Similarly, we observed the same trend with the doxycycline MIC. The doxycycline MIC exhibited a significant decrease in the Δ*sa00565* strain (0.125 mg/L) as compared to the WT strain (1 mg/L). The *sa00565* overexpression strain further exhibited a higher MIC of 12 mg/L, which was a 12-fold increase as compared to the WT strain (Fig. 3). These findings provided substantive evidence supporting the pivotal role of the *sa00565* gene in tetracycline and doxycycline resistance in *S. aureus*.

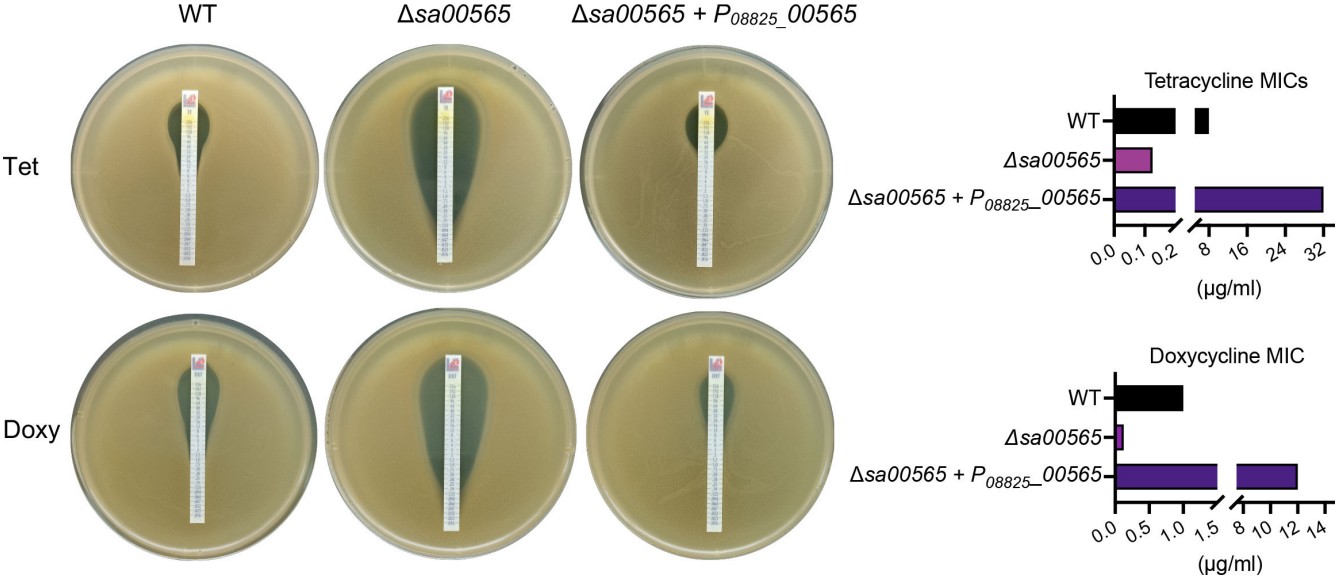

**FIG 3** MICs of Tetracycline (Tet) and Doxycycline (Doxy) conducted on *S. aureus* WT, Δ*sa00565*, and Δ*sa00565* overexpression (Δ*sa00565* + P$_{08825\_}$00565) strains. The assay was performed using the MIC gradient strips on TSA plates in compliance with CLSI guidelines. The MICs of Tet and Doxy for each strain were determined by reading the plate and presenting the data as histograms in the right panel.

## The efflux mechanism facilitated the role of SA00565 in conferring tetracycline resistance

The deletion of the *sa00565* gene exhibited augmented sensitivity of *S. aureus* towards tetracycline. Tetracycline is renowned for its antimicrobial properties, achieved by effectively targeting the bacterial intracellular ribosome server (13–15). We postulated that SA00565 functioned as an efflux pump that facilitated the expulsion of tetracycline from inside the bacterial cells to the external environment. To test this hypothesis, the *S. aureus* WT and the Δ*sa00565* strains were exposed to 5 µg/mL of tetracycline, and the concentration kinetics of tetracycline within the bacterial cells of each strain were determined through enzyme-linked immunosorbent assay (ELISA). The outcome revealed that the concentration of intracellular tetracycline in the Δ*sa00565* mutant was two- to threefold greater than that observed in the WT strain (Fig. 4A). This

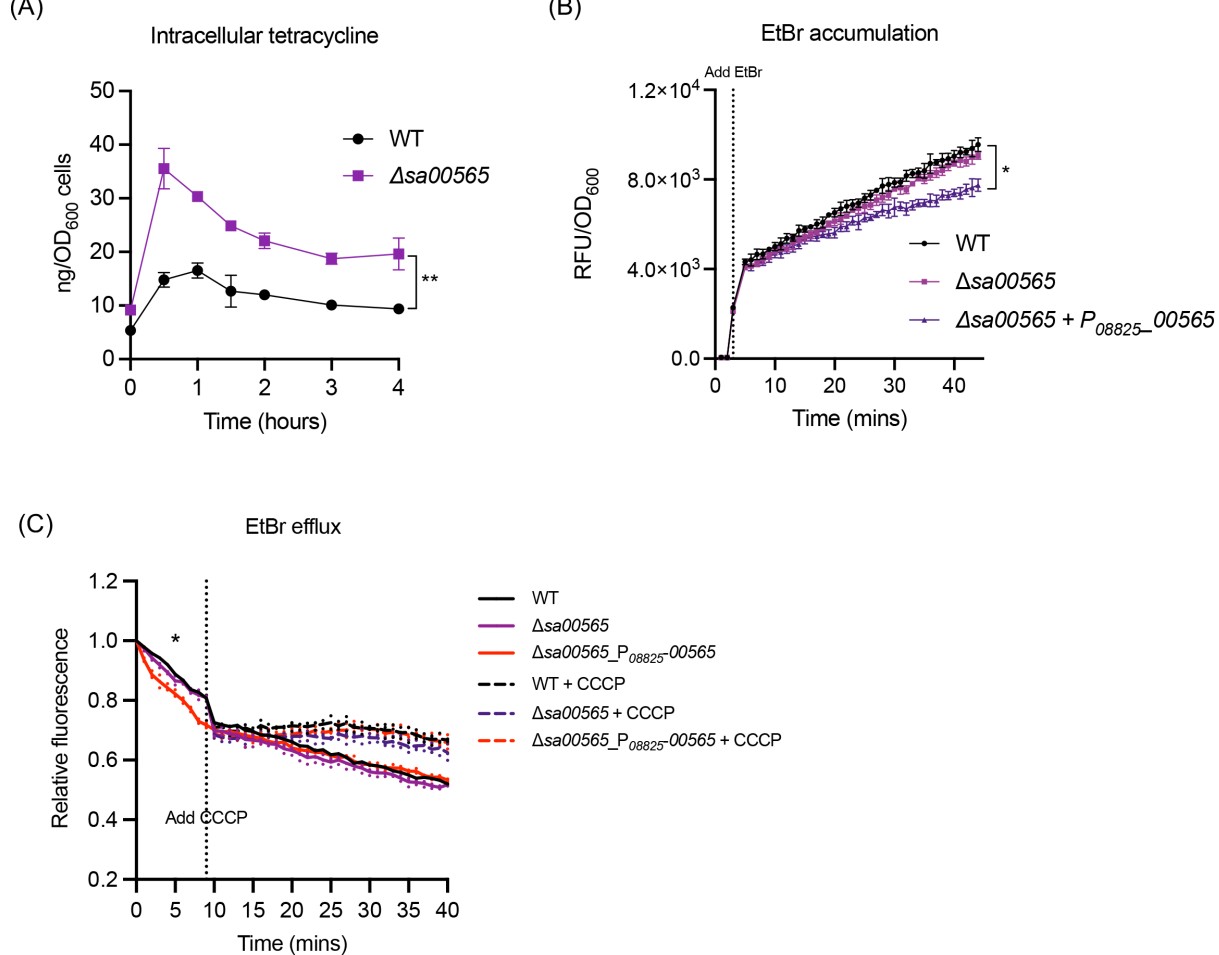

**FIG 4** Intracellular tetracycline and EtBr accumulation and efflux assays. (A) *S. aureus* WT and the Δ*sa00565* mutant were treated with a final concentration of 5 µg/mL tetracycline. At various time points (0, 0.5, 1, 1.5, 2, 3, and 4 h) after treatment, the cells were collected and resuspended in lysis buffer. The concentration of tetracycline in the cell lysate was quantified using ELISA. The tetracycline concentrations in the WT and Δ*sa00565* groups were compared and analyzed during the indicated period (**$P \leq 0.01$ by paired *t* test). (B) EtBr accumulation assay. *S. aureus* cells were exposed to 4 µg/mL EtBr within a 96-well plate. Fluorescence and $OD_{600}$ values were measured for each well, and RFUs were calculated by normalizing fluorescence data to the respective $OD_{600}$ values. The dotted line on the graph indicates the precise moment of EtBr addition. Statistical analyses were executed in reference to the WT (*$P \leq 0.05$ by unpaired *t* test). (C) EtBr efflux assay. Following a 10-min incubation period of bacterial cells with EtBr, extracellular EtBr was eliminated through centrifugation. The resulting cells were then resuspended in PBS and their associated RFUs were monitored for a subsequent period of 10 min. Subsequently, each well was treated with the efflux pump inhibitor CCCP, and RFUs were monitored for an additional 30 min. Significance levels were evaluated by conducting unpaired *t* tests in reference to the WT (*$P \leq 0.05$ by unpaired *t* test).

result supported the assertion that SA00565 worked as an efflux pump and conferred tetracycline resistance to *S. aureus* through the elimination of intracellular tetracycline.

To further confirm the efflux activity of SA00565, EtBr accumulation tests were conducted on WT, Δ*sa00565*, and *sa00565* overexpression strains. The strains were exposed to EtBr, and their fluorescence intensities were monitored over time. As depicted, a time-dependent fluorescence increase was observed for all strains, with no significant differences between WT and Δ*sa00565* knockout strains (Fig. 4B). Conversely, the SA00565-overexpressing strain exhibited substantially less fluorescence increase compared to both the WT and Δ*sa00565* strains. This discrepancy suggested the existence of a more robust efflux activity in the SA00565-overexpressing strain when compared to the WT and Δ*sa00565* strains.

In the EtBr efflux assay, the fluorescent values of the *sa00565*-overexpressing strain declined at a significantly faster rate than those of WT and Δ*sa00565* strains during the period of EtBr extrusion (Fig. 4C). These findings suggested that overexpression of the *sa00565* gene substantially increased the efflux activity of *S. aureus*. Upon treatment with the protonophore carbonyl cyanide-chlorophenylhydrazone (CCCP), which disrupts the electrochemical potential of $H^+$ across the cytoplasmic membrane and thereby serves as a driving force for secondary transporters, the fluorescence of all tested strains ceased decreasing due to the dissipation of the proton gradient across the membrane (Fig. 4C). This observation provided compelling evidence suggestive of the $H^+$-dependent activity of SA00565. Collectively, these data established SA00565 as an active efflux pump that confers tetracycline resistance in *S. aureus* through the extrusion of intracellular tetracycline.

## DISCUSSION

In this investigation, the role of the *sa00565* gene in the mediation of antimicrobial resistance of *S. aureus* USA300 was examined. Through prediction analysis, *sa00565* was identified as encoding an efflux pump that is part of the DMT superfamily, which is a diverse group of transporters characterized by 14 subfamilies classified according to sequence similarity and transmembrane topology (11). Pertinent to this study, the DMT transporters found in each subfamily carry out conserved functions, such as members of the GRP and RarD subfamilies that are involved in sugar transportation, while the SMR and DME subfamilies transport drugs. To gain insight into the function of the efflux pump SA00565, a phylogenetic tree of SA00565 in the DMT superfamily proteins was constructed. The results revealed a close relationship between SA00565 and the DME subfamily, which led to the supposition that SA00565 might function as a drug efflux pump in *S. aureus*. This hypothesis was subsequently confirmed by the antibiotic susceptibility test and MIC assay, which indicated that SA00565 mediated tetracycline and doxycycline resistance in *S. aureus*.

Bacterial resistance to tetracycline is currently accounted for by three recognized mechanisms, namely active efflux, ribosomal protection, and enzymatic inactivation of tetracycline (16). Multiple studies have demonstrated the involvement of various transporters in *S. aureus*, such as SA09310, NorB, Tet38, and MepA, in conferring tetracycline resistance via active efflux mechanisms, resulting in the extrusion of tetracycline from the intracellular milieu into the external environment (6, 17–19). Notably, a substantial reduction in intracellular tetracycline levels was observed in the Δ*sa00565* strain relative to the *S. aureus* WT upon exposure to tetracycline, thus implying the involvement of the efflux pump SA00565 in conferring tetracycline resistance via a shared active efflux mechanism. This efflux activity of SA00565 was further verified by EtBr accumulation and efflux assays. Through the aggregation of previously established knowledge regarding tetracycline efflux pumps, it becomes apparent that there are various efflux pumps that can facilitate tetracycline resistance in *S. aureus* via a common mechanism. The rationale for the extensive employment of efflux pumps for tetracycline resistance in *S. aureus* remains elusive and warrants further exploration. Moreover, the

possible existence of additional functionalities of these pumps, beyond the efflux of tetracyclines, deserves further investigation.

Despite the fact that SA00565 enhanced the ability of *S. aureus* to resist against tetracycline and doxycycline, the deletion of *sa00565* gene did not result in any significant increase in the susceptibility of *S. aureus* to minocycline and tigecycline, both of which are tetracycline antibiotics (Fig. S2). After examining the structures of tetracycline, doxycycline, minocycline, and tigecycline, we noted that doxycycline shares the closest resemblance with tetracycline, followed by minocycline and tigecycline, which have undergone more modifications to their tetracycline backbone. These observations imply that SA00565 acts in recognition and extrusion of tetracycline with a structural specificity. Consequently, the recognition and extrusion by SA00565 of substrates that are structured differently from the original tetracycline become more difficult. Thus, the variant structure of minocycline and tigecycline might explain their evasion from recognition and extrusion by SA00565 compared to tetracycline.

In conclusion, we have successfully identified a novel DME subfamily transporter belonging to the DMT superfamily in *S. aureus*. Our findings unequivocally confirm the crucial role of SA00565 in mediating tetracycline efflux, which significantly contributes to tetracycline resistance in *S. aureus*.

## MATERIALS AND METHODS

### Bacterial strains and growth conditions

The *S. aureus* USA300_FPR3757 strain served as the sole source for the generation of all *S. aureus* mutants in this study. Plasmids utilized for *S. aureus* USA300 transformation underwent modifications by *S. aureus* RN4220. Electroporation, following previously published procedures (20), was employed to generate all *S. aureus* transformants. All *S. aureus* strains were cultivated by shaking in tryptic soy broth (TSB) at 220 rpm or on TSB agar plates (TSA) incubated at 37°C. For plasmid cloning, DH5α was used, and it was cultured in Luria-Bertani (LB) broth continuously shaken at 220 rpm or on LB agar plates at 37°C. Antibiotics were included at the following concentrations where specified: 100 µg/mL ampicillin and 25 µg/mL chloramphenicol.

### Phylogenetic analysis of SA00565 in DMT superfamily

The transmembrane helices of SA00565 protein were inferred using the TMHMM 2.0 online tool (http://www.cbs.dtu.dk/services/TMHMM/), while the classification of SA00565 transporters was ascertained through reference to the transporter classification database (21). To construct the phylogenetic tree of SA00565 within the DMT protein family, we collected at least two or more protein sequences from each of the 14 classified subgroups in the DMT family, namely SMR with 4 transmembrane segments (TMS), bacterial/archaeal transporter with 5 TMS (BAT), drug/metabolite exporter with 10 TMS (DME), plant drug/metabolite exporter (P-DME), glucose/ribose porter (GPR), L-rhamnose transporter (RhaT), Chloramphenicol-sensitivity protein (RarD), *Caenorhabditis elegans* ORF (CEO), Triose phosphate transporter (TPT), UDP-*N*-acetylglucosamine: UMP antiporter (UAA), UDP-galactose: UMP antiporter (UGA), CMP-sialate: CMP antiporter (CSA), GDP-mannose: GMP antiporter (GMA), and Plant organocation permease (POP) (11), which were acquired from UniProt. We performed multiple sequence alignments between SA00565 and DMT proteins using ClustalW (22), a commonly used algorithm for multiple sequence alignment. The resulting phylogenetic tree was generated using the neighbor-joining method and the Jukes-Cantor protein distance model from the multiple alignments, as implemented in CLC Main Workbench (Qiagen, USA).

### Generation of *sa00565* knockout and complementary overexpression strains

To generate a deletion mutant of the *sa00565* gene, we performed PCR using primer pairs QL1195/QL1196 and QL1197/QL1198 to amplify DNA fragments approximately

1 kb in length flanking the *sa00565* gene. All primers used in this study were listed in Table S1. The two amplified fragments were joined using fusion PCR, facilitated by their overlap sequences. Subsequently, the fused fragment was integrated into the *S. aureus-Escherichia coli* shuttle vector pBT2 (23) via recombination, after the vector had been linearized via EcoRI and SalI. By transforming the resulting plasmid into *S. aureus* USA300 WT, we obtained the *sa00565* gene deletion mutant (Δ*sa00565*) through two rounds of recombination, as previously described (6). The mutant was confirmed by PCR using primers QL1199 and QL1200.

To generate the complementary repaired strain (Δ*sa00565* + P$_{08825}$_*00565*) of Δ*sa00565*, we employed a modified pQLV1025 gene expression plasmid with the strong constitutive promoter P$_{08825}$ from *S. aureus* (24). In brief, the original tetracycline selection marker from pQLV1025 was removed by PCR using pQLV1025 as a DNA template with primers QL1496 and QL1497. The resulting fragment was self-ligated using homologous sequences introduced in the primers. Subsequently, the *sa00565* gene was amplified using primers QL1347 and QL1348 and cloned into the modified expression vector between the NdeI and XhoI sites. The elevated expression level of the *sa00565* gene in the repaired strain was confirmed via RT-qPCR, employing primer pairs QL1349/QL1350 for the *sa00565* gene and QL0152/QL0153 for the *16S* rRNA gene, as previously described (6). Fold changes in expression level relative to the WT were determined by employing the comparative Ct (ΔΔCt) method.

## Disk diffusion and MIC assays

The disk diffusion assay was performed as previously described (6). Briefly, bacterial cells in mid-exponential phase were spread on TSA plates and exposed to 6 mm diameter antibiotic disks with distinct antibiotics (Table S2) purchased from Microbial Regent Company (Hangzhou Microbial, Hangzhou, China). After incubation for 12 h, zones of inhibition were recorded. Sensitivities were assessed by measuring the diameters of the zones of growth inhibition in three independent assays.

For MICs test, we employed the MIC gradient strips (Abruzzi, Italy). The bacteria-spread plates were prepared following the previously described protocol. The MIC gradient strips, containing gradients of tetracycline or doxycycline from 0.016 to 256 mg/L. The strips were placed onto the TSA surface by using forceps. After incubating the plates at 37°C for 24 h, the MIC value was interpreted by observing the symmetrical inhibition ellipse on the plate, as per guidelines provided by the Clinical and Laboratory Standards Institute (CLSI).

## Measurement of intracellular tetracycline by ELISA

A day culture of *S. aureus* was started with 100 µL of overnight culture incubated in 10 mL of TSB medium for 2 h at 37°C with continuous shaking at 220 rpm. Then, 5 µg/mL of tetracycline (Merk, Germany) was added to the culture. Bacterial cells were collected at time points 0.5, 1, 1.5, 2, 3, 4, or 5 h after tetracycline addition. The cells were washed three times with phosphate-buffered saline (PBS) and resuspended in lysis buffer (20 mM Tris-HCl, pH 8.0; 2 mM sodium EDTA; 1.2% Triton X-100) with an OD$_{600}$ value of 1.0 (5 × 10$^8$ CFU/mL). About 50 µg/mL lysostaphin (Sangon, China) was added to lyse the cells, and the intracellular tetracycline released was quantified using a tetracycline ELISA kit (Ruixin Biotech, Quanzhou, China) according to the manufacturer's instructions. Briefly, the supernatant of the lysate was mixed with anti-tetracycline antibody and added to a pre-coated tetracycline well on a 96-well plate. The mixture was incubated at 37°C for 30 min, washed with PBS five times, and incubated with substrate solution for 15 min. The reaction was stopped with stop solution, and the absorbance value of each well at 450 nm was recorded. A standard curve was generated using absorbance values and concentrations of tetracycline standard samples. The concentration of tetracycline from cell lysis was calculated using the standard curve equation based on the absorbance value.

## Ethidium bromide (EtBr) accumulation and efflux assays

*S. aureus* cells were collected from the mid-exponential phase by centrifugation. The resulting pellet was washed with PBS three times, before being resuspended in PBS, containing 10 mM glucose, to achieve an optical density of $OD_{600} = 0.4$. Next, 100 µL of the cell suspension was dispensed in triplicate into wells of a 96-well black plate, with a clear bottom. The plate was used to measure the baseline cellular fluorescence for 2 min. Following this, EtBr (Sangon, China) was added to each well, to achieve a final concentration of 4 µg/mL, and the fluorescence was measured every 30 s for 40 min, using a Synergy H1 plate reader (Bio-Tek, USA), set at emission and excitation wavelengths of 580 and 500 nm, respectively.

The EtBr efflux assay was carried out by incubating *S. aureus* cells with EtBr at a final concentration of 20 µg/mL, for a duration of 10 min. Thereafter, the extracellular EtBr was eliminated through a centrifugation step followed by resuspension of the cells in fresh PBS, and the fluorescence of each strain was monitored using the method described above. Upon completion of the initial 10 minute fluorescence monitoring, the efflux pump inhibitor, CCCP (Sangon, China), was added to each well at a final concentration of 100 µM, and the fluorescence was monitored for an additional 30 min.

## ACKNOWLEDGMENTS

The authors would like to recognize the staff at the West China Biopharmaceutical Research Institute for their vital assistance in this study.

This work was supported by the Science Fund for Distinguished Young Scholars of Sichuan Province (2022NSFSC1682), the National Science Fund for Distinguished Young Scholars (32000094), the China Postdoctoral Science Foundation (2021M692311), the Post-Doctor Research Project, West China Hospital, Sichuan University (20HXBH017), and the 1·3·5 Project for Disciplines of Excellence, West China Hospital, Sichuan University (ZYXY21004).

## AUTHOR AFFILIATIONS

[1]West China Biopharmaceutical Research Institute, West China Hospital, Sichuan University, Chengdu, China
[2]Department of Microbiology and Biochemical Pharmacy, National Engineering Research Center of Immunological Products, College of Pharmacy, Third Military Medical University, Chongqing, China

## AUTHOR ORCIDs

Qiang Liu [ID] http://orcid.org/0000-0002-2527-6594

## FUNDING

| Funder | Grant(s) | Author(s) |
| --- | --- | --- |
| SPDST | Basic Research Programs of Sichuan Province | 2022NSFSC1682 | Qiang Liu |
| MOST | NSFC | China National Funds for Distinguished Young Scientists | 32000094 | Qiang Liu |
| SCU | West China Hospital, Sichuan University (WCH) | ZYXY21004 | Yun Shi |

## AUTHOR CONTRIBUTIONS

Daiyu Li, Data curation, Investigation, Methodology, Validation | Yan Ge, Investigation, Methodology | Ning Wang, Investigation, Methodology | Yun Shi, Data curation, Funding acquisition, Validation | Gang Guo, Formal analysis, Supervision | Jing Zhang, Investigation, Methodology | Quanming Zou, Data curation, Resources, Supervision | Qiang Liu, Conceptualization, Data curation, Formal analysis, Funding acquisition, Investiga-

tion, Methodology, Project administration, Resources, Software, Supervision, Validation, Visualization, Writing – original draft, Writing – review and editing

## ADDITIONAL FILES

The following material is available online.

### Supplemental Material

**Supplemental material (Spectrum00111-24-s0001.pdf).** Fig. S1 and S2; Tables S1 and S2.

### Open Peer Review

**PEER REVIEW HISTORY (review-history.pdf).** An accounting of the reviewer comments and feedback.

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
