## [Reviewer comments · Microbiology Spectrum]

Microbiology Spectrum

A Novel Member of Drug/Metabolite Transporter (DMT) Family Efflux Pump, SA00565, Contributes to Tetracycline Antibiotics Resistance in *Staphylococcus aureus* USA300

Daiyu Li, Yan Ge, Ning Wang, Yun Shi, Gang Guo, Jing Zhang, Quanming Zou, and Qiang Liu

Corresponding Author(s): Qiang Liu, Sichuan University West China Hospital

Review Timeline:

Submission Date:	January 12, 2024
Editorial Decision:	March 4, 2024
Revision Received:	March 18, 2024
Accepted:	March 28, 2024

Editor: Emily Weinert

Reviewer(s): The reviewers have opted to remain anonymous.

Transaction Report:

DOI: <https://doi.org/10.1128/spectrum.00111-24>

Re: Spectrum00111-24 (A Novel Member of Drug/Metabolite Transporter (DMT) Family Efflux Pump, SA00565, Contributes to Antibiotic Resistance in *Staphylococcus aureus* USA300)

Dear Dr. Liu:

Thank you for the privilege of reviewing your work. Below you will find my comments, instructions from the Spectrum editorial office, and the reviewer comments.

Revision Guidelines

Sincerely,
Emily Weinert
Editor
Microbiology Spectrum

Reviewer #1 (Comments for the Author):

The authors emphasize the function of the Drug/Metabolite Transporter (DMT) Family Efflux Pump, SA00565 transporter in tetracycline resistance in *S. aureus*. The manuscript is moderately good and requires modifications.

Reviewer #2 (Comments for the Author):

Spectrum 00111-24 studies the role of SA00565 in *S. aureus* physiology. Using a bioinformatic approach, the authors identified that the gene could encode for an efflux pump, and through genetic knock-out and complementation studies they confirmed that the expressed gene plays a role in tetracycline and doxycycline efflux.

Minor Comments:

- 1) Title: The use of "Antibiotic Resistance" in the title is too broad. The authors show it is limited to tetracycline and doxycycline.
- 2) L23, L84: similar as above "multidrug" is too broad; delete
- 3) L116" replace "triggered" with "showed"
- 4) L118: replace "restored" with "reverted"
- 5) L126, L271: "E-Test" is a trademark from bioMerieux, and those were not used in this study. Use "MIC gradient strips" instead.
- 5) L140-141: "Tetracycline is a compound that acclaimed for its antimicrobial activity by targeting the server of bacterial intracellular ribosome (16-18)." Please rephrase, unclear what is meant by this.
- 6) Fig 2B: please use log scale for Y-axis.
- 7) Fig 3: can be moved to supplementary data.

The authors emphasize the function of the Drug/Metabolite Transporter (DMT) Family Efflux Pump, SA00565 transporter in tetracycline resistance in *S. aureus*. The manuscript is moderately good and requires modifications.

1. Body of work:

Line 27: "mutant exhibited significantly reduced resistance" suggest to write "exhibited reduced resistance"

Line 38,44: *Staphylococcus aureus*" suggest to italicize your text.

Line 53-54: causing numerous diseases in human" suggest to write the name of diseases.

Line 55-60: Add the most recent references.

1) Akshay SD, Deekshit VK, Mohan Raj J, Maiti B. Outer Membrane Proteins and Efflux Pumps Mediated Multi-Drug Resistance in *Salmonella*: Rising Threat to Antimicrobial Therapy. *ACS Infectious Diseases*. 2023 Nov 1;9(11):2072-92.

2) Li D, Ge Y, Wang N, Shi Y, Guo G, Zou Q, Liu Q. Identification and Characterization of a Novel Major Facilitator Superfamily Efflux Pump, SA09310, Mediating Tetracycline Resistance in *Staphylococcus aureus*. *Antimicrobial Agents and Chemotherapy*. 2023 Apr 18;67(4):e01696-22.

3) Huang H, Wan P, Luo X, Lu Y, Li X, Xiong W, Zeng Z. Tigecycline Resistance-Associated Mutations in the MepA Efflux Pump in *Staphylococcus aureus*. *Microbiology Spectrum*. 2023 Aug 17;11(4):e00634-23.5. Line 82: The authors have mentioned azathioprine (50 mg/d) for ongoing treatment; explain the use of azathioprine for inflammatory bowel diseases. Provide an appropriate reason.

Line 70,75 & 307: "Aforementioned" should be replaced with an appropriate name.

Line 77: "staphylococci" suggest to write "Staphylococci"

2. In the methodology section, Line 259: Mention the RT-PCR methodology as well as the name of the statistical analysis. The author only commented that the expression was confirmed using RT-PCR.

3. In the methodology section, Disk diffusion and MIC assays: The author mentioned the disk diffusion test used in the investigation. It is of utmost importance to understand the type of diffusion method employed. Give a justification for using TSA agar for the disk diffusion method. Provide standard

guideline references (CLSI or EUCAST) for using TSA for the disk diffusion technique or suggest 2-3 recent references for using the TSA agar for the test.

4. In the manuscript, chemical and equipment manufacturer references must be provided throughout the paper. Suggest writing the company's name and country of origin.

5. Supplementary Fig. 1. should have the proper statistical analysis. What is Δ sa00565 + P08825 _00565 significance when compared with WT? Also, provide the explanations for having 900-fold upregulation. Suggest references for having a similar kind of expression in a higher fold.

6. What are the prospective outcomes of this study? How will this study aid in preventing antibiotic resistance?

7. The manuscript contains quite a few typographical errors.

Reviewer #1 (Comments for the Author):

The authors emphasize the function of the Drug/Metabolite Transporter (DMT) Family Efflux Pump, SA00565 transporter in tetracycline resistance in *S. aureus*. The manuscript is moderately good and requires modifications.

1. Body of work:

Line 27: “mutant exhibited significantly reduced resistance” suggest to write “exhibited reduced resistance”

We acknowledge the suggestion and have modified the text accordingly to “exhibited reduced resistance.”

Line 38,44: “*Staphylococcus aureus*” suggest to italicize your text.

We have italicized “*Staphylococcus aureus*” as suggested.

Line 53-54: causing numerous diseases in human” suggest to write the name of diseases.

As suggested, we have delineated the diseases attributed to *S. aureus*. As below:

“*Staphylococcus aureus* is a gram-positive bacterial species responsible for causing bacteraemia, endocarditis, skin and soft tissue infections, bone and joint infections and hospital-acquired infections”

Line 55-60: Add the most recent references.

1) Akshay SD, Deekshit VK, Mohan Raj J, Maiti B. Outer Membrane Proteins and Efflux Pumps Mediated Multi-Drug Resistance in *Salmonella*: Rising Threat to Antimicrobial Therapy. ACS Infectious Diseases. 2023 Nov 1;9(11):2072-92.

2) Li D, Ge Y, Wang N, Shi Y, Guo G, Zou Q, Liu Q. Identification and Characterization of a Novel Major Facilitator Superfamily Efflux Pump, SA09310, Mediating Tetracycline Resistance in *Staphylococcus aureus*. Antimicrobial Agents and Chemotherapy. 2023 Apr 18;67(4):e01696-22.

3) Huang H, Wan P, Luo X, Lu Y, Li X, Xiong W, Zeng Z. Tigecycline Resistance-Associated Mutations in the MepA Efflux Pump in *Staphylococcus aureus*. Microbiology Spectrum. 2023 Aug 17;11(4):e00634-23.5.

We have included the most recent references as you provided.

Line 70,75 & 307: "Aforementioned" should be replaced with an appropriate name.

The "Aforementioned" in the manuscript was replaced with appropriate word.

Line 77: " staphylococci” suggest to write “Staphylococci”

We have corrected "staphylococci" to "Staphylococci"

2. In the methodology section, Line 259: Mention the RT-PCR methodology as well as the name of the statistical analysis. The author only commented that the expression was confirmed using RT-PCR.

The RT-PCR methodology and the name of the statistical analysis was mentioned in the revised manuscript.

3. In the methodology section, Disk diffusion and MIC assays: The author mentioned the disk diffusion

test used in the investigation. It is of utmost importance to understand the type of diffusion method employed. Give a justification for using TSA agar for the disk diffusion method. Provide standard
As you mentioned, the selection of agar medium significantly influences the precision of antimicrobial susceptibility testing through disk diffusion.

TSA is commonly used for disk diffusion assays due to its ability to support the growth of a wide range of microorganisms, including both fastidious and non-fastidious bacteria. Moreover, TSA agar yields reliable outcomes and is endorsed by recognized standards such as CLSI (Clinical and Laboratory Standards Institute) or EUCAST (European Committee on Antimicrobial Susceptibility Testing) for disk diffusion testing.

4. In the manuscript, chemical and equipment manufacturer references must be provided throughout the paper. Suggest writing the company's name and country of origin.

Thank you for your reminder. These missing information was provided in the revised manuscript.

5. Supplementary Fig. 1. should have the proper statistical analysis. What is Δ sa00565 + P08825_00565 significance when compared with WT? Also, provide the explanations for having 900-fold upregulation. Suggest references for having a similar kind of expression in a higher fold.

Statistical analysis was conducted on the expression levels between the wild-type (WT) and sa00565-overexpressing strains, as depicted in Supplementary Fig. 1. The observed 900-fold upregulation of the sa00565 gene in the overexpression strain resulted from the utilization of a potent constitutive promoter, as opposed to its natural counterpart. These information was incorporated into the revised manuscript.

6. What are the prospective outcomes of this study? How will this study aid in preventing antibiotic resistance?

By identifying and characterizing SA00565 and elucidating its role in antibiotic resistance, the study provides valuable insights into the mechanisms driving multidrug resistance in *S. aureus*.

Understanding the specific mechanisms of resistance, such as efflux pump activity, can inform the development of new treatment strategies, including the design of efflux pump inhibitors or alternative antibiotics that are less susceptible to efflux-mediated resistance.

7. The manuscript contains quite a few typographical errors.

We have thoroughly proofread the manuscript to correct typographical errors

Reviewer #2 (Comments for the Author):

Spectrum 00111-24 studies the role of SA00565 in *S. aureus* physiology. Using a bioinformatic approach, the authors identified that the gene could encode for an efflux pump, and through genetic knock-out and complementation studies they confirmed that the expressed gene plays a role in tetracycline and doxycycline efflux.

Minor Comments:

1) Title: The use of "Antibiotic Resistance" in the title is too broad. The authors show it is limited to tetracycline and doxycycline.

We have changed the "Antibiotic Resistance" to "Tetracycline Antibiotics Resistance" in title.

2) L23, L84: similar as above "multidrug" is too broad; delete

As you suggested, the "multidrug" in line 23 and line 84 have been deleted.

3) L116" replace "triggered" with "showed"

We have corrected "triggered" to "showed".

4) L118: replace "restored" with "reverted"

We have replaced "restored " with "reverted".

5) L126, L271: "E-Test" is a trademark from bioMerieux, and those were not used in this study. Use "MIC gradient strips" instead.

Thanks for the reminder. We have made the changes.

5) L140-141: "Tetracycline is a compound that acclaimed for its antimicrobial activity by targeting the server of bacterial intracellular ribosome (16-18)." Please rephrase, unclear what is meant by this.

The sentence has been rephrased in the revised manuscript as below :

"Tetracycline is renowned for its antimicrobial properties, achieved by effectively targeting the bacterial intracellular ribosome server."

6) Fig 2B: please use log scale for Y-axis.

The Y-axis in Figure 2B represent the OD₆₀₀ value of the bacterial culture. We believe it's more appropriate to retain the original OD₆₀₀ values.

7) Fig 3: can be moved to supplementary data.

Thank you for the suggestion. Figure 3 shows the MICs of tetracycline doxycycline for each strain. Although the disk diffusion assay in Figure 2 shows the difference in sensitivity of each strain to different antibiotics, Figure 3 shows the quantitative detection of antibiotic sensitivity. Therefore, we believe that this is also an important indicator to evaluate SA00565 efflorescent pump-mediated drug resistance, so we think that Figure 3 can be retained in the body.

Re: Spectrum00111-24R1 (A Novel Member of Drug/Metabolite Transporter (DMT) Family Efflux Pump, SA00565, Contributes to Tetracycline Antibiotics Resistance in *Staphylococcus aureus* USA300)

Dear Dr. Liu:

Your manuscript has been accepted, and I am forwarding it to the ASM production staff for publication. Your paper will first be checked to make sure all elements meet the technical requirements. ASM staff will contact you if anything needs to be revised before copyediting and production can begin. Otherwise, you will be notified when your proofs are ready to be viewed.

Sincerely,
Emily Weinert
Editor
Microbiology Spectrum